Interactions between two functionally distinct aquatic invertebrate herbivores complicate ecosystem- and population-level resilience

Werba Jo A. jo.werba@gmail.com
Phong Alexander C.
Brar Lakhdeep
Frempong-Manso Acacia
Oware Ofure Vanessa
Kolasa Jurek
Department of Biology, McMaster University , Hamilton , Ontario , Canada
Silva Daniel
Electronic publication date: 2022 Oct 7
Publication date: 2022
Volume: 10
Electronic Location ID: e14103
Received 2022 Feb 28; Accepted 2022 Sep 1
Copyright: ©2022 Werba et al.
Copyright year: 2022
Copyright holder: Werba et al.
License: This is an open access article distributed under the terms of the Creative Commons Attribution License, which permits unrestricted use, distribution, reproduction and adaptation in any medium and for any purpose provided that it is properly attributed. For attribution, the original author(s), title, publication source (PeerJ) and either DOI or URL of the article must be cited.
License URL: https://creativecommons.org/licenses/by/4.0/

Keywords: Functional traits, Freshwater, Eutrophication, Perturbation, Resilience, Physa, Daphnia magna

Funding: Natural Sciences and Engineering Research Council of Canada (NSERC) 10531314 This work is supported by the Natural Sciences and Engineering Research Council of Canada (NSERC) 10531314 to Jurek Kolasa. The funders had no role in study design, data collection and analysis, decision to publish, or preparation of the manuscript.

==============================
Resilience, the capacity for a system to bounce-back after a perturbation, is critical for conservation and restoration efforts. Different functional traits have differential effects on system-level resilience. We test this experimentally in a lab system consisting of algae consumed by zooplankton, snails, or both, using an eutrophication event as a perturbation. We examined seston settlement load, chlorophyll-a and ammonium concentration as gauges of resilience. We find that Daphnia magna increased our measures of resilience. But this effect is not consistent across ecosystem measures; in fact, D. magna increased the difference between disturbed and undisturbed treatments in seston settlement loads. We have some evidence of shifting reproductive strategy in response to perturbation in D. magna and in the presence of Physa sp. These shifts correspond with altered population levels in D. magna, suggesting feedback loops between the herbivore species. While these results suggest only an ambiguous connection between functional traits to ecosystem resilience, they point to the difficulties in establishing such a link: indirect effects of one species on reproduction of another and different scales of response among components of the system, are just two examples that may compromise the power of simple predictions.

Introduction

Resilience is an ecosystem’s capacity to withstand change while maintaining processes and structures (Chaffin et al., 2016). Understanding resilience and predicting when a system may be resilient is important to conservation for prioritizing areas to preserve and maximizing the natural world’s capacity to “bounce back”, or resist change. However, defining and measuring resilience has been challenging. Resilience can refer to the degree of perturbation a system can sustain (Gunderson, 2000; Ludwig, Walker & Holling, 1997), a length of time until the systems returns to a pre-perturbation state (elasticity e.g., sensu Hodgson, McDonald & Hosken, 2015), or a combination of both (Hodgson, McDonald & Hosken, 2015; Yeung & Richardson, 2016; Hodgson, McDonald & Hosken, 2016; Côté & Darling, 2010). Despite the relatively simple idea, the variety of definitions of resilience and the difficulty in measuring resilience has led to inconclusive and often contradictory findings about what makes a system more likely to be resilient (Todman et al., 2016; Ingrisch & Bahn, 2018). Because resilience is often a conservation goal, the contradictory evidence and vague definitions leaves policy and planning difficult (Newton, 2016).

While there are many works on the theoretical underpinnings of mechanisms that increase resilience, these works have yielded mixed results. Resilience is thought to increase by a variety of mechanisms such as: increased biodiversity (e.g. Allan et al., 2011; Tilman & Downing, 1994), changes in complex pathways (e.g.changes in food chain dynamics Downing & Leibold, 2010), insurance (Mori, Furukawa & Sasaki, 2013) or by increasing the likelihood of having a critical species present (the sampling effect) (e.g. Steiner et al., 2006). However, counter-examples are also common for each work that supports a particular mechanism that increases resilience. For example, Guelzow et al. (2017) failed to find increases in resilience with increases in biodiversity, and Timóteo et al. (2016) failed to find a sampling effect and instead found an insurance effect. Vertical (trophic) effects such as the food chain dynamics or nutrient cycling have received less attention, yet may be more critical as they are the core scaffolding for ecosystem persistence (Abdala-Roberts et al., 2019). Overall, it is reasonable to postulate that mechanisms that increase resilience are system dependent.

If this postulate is correct, functional traits should be a critical factor for understanding resilience instead of taxonomic or species richness (e.g., Krztoń, Kosiba & Wilk-Woźniak, 2022). Functional trait-based approaches can meaningfully reduce complexity while providing reasonable predictions across environmental gradients for both community composition and ecosystem functions (e.g., Abonyi, Horváth & Ptacnik, 2018; Bremner, 2008; Cadotte, Carscadden & Mirotchnick, 2011; Cardinale, Nelson & Palmer, 2000). By focusing on mechanisms, ecosystem functions as diverse as productivity (e.g., Cardinale, Nelson & Palmer, 2000; van der Sande et al., 2018), nitrogen cycling (e.g., Craine et al., 2002) and resilience (e.g., Hu et al., 2022; Peterson, Allen & Holling, 1998) have been successfully linked to species traits. Since different functional traits impact different parts of the ecosystem, individual functional groups should respond differently to the same or different disturbances. Understanding which and what combination of functional groups is necessary for resilience in the face of particular disturbance is the first step to linking system structure to its resilience in general.

Eutrophication of freshwaters is a common threat to biodiversity and the functioning of aquatic communities (Fetahi, 2019; Geng et al., 2022; Saunders, Meeuwig & Vincent, 2002; Carpenter, Ludwig & Brock, 1999). Eutrophication generally occurs from run-off from agricultural or mining lands. This run-off leads to algal blooms that can be toxic, potentially so excessive as to drastically deplete oxygen in the water body (Smith, 1998), and even changes evolutionary trajectories of species (Brede et al., 2009). Reversing eutrophication can be challenging because removing nutrients alone is sometimes insufficient to reverse all effects (Carpenter, Ludwig & Brock, 1999; Brede et al., 2009). Thus, understanding which aspects of a system will increase resilience to eutrophication is useful. Because different functional groups can have differential resiliency to perturbation (e.g., Karp et al., 2011) or impart resiliency to the whole system in different ways, a focus on such groups may help. Specifically, this may be crucial for nutrient cycling where individual functional groups may differently affect rates of nutrients processing and in different parts of the nutrient cycle (e.g., decomposition, herbivory) (Hulot, Lacroix & Loreau, 2014).

Scrapers, such as the snail Physa sp., and filter feeders such as the Cladoceran, Daphnia magna have well-known effects on aquatic systems. For example, snails alter multiple pieces of a freshwater ecosystem. Snails alter fish and even bird communities (Gilioli et al., 2017), periphyton abundance and diversity (Swamikannu & Hoagland, 1989), are major decomposers (Brady & Turner, 2010) and are likely important for nutrient cycling (even contributing up to 2/3 of all ammonium in a system) (Hall Jr, Tank & Dybdahl, 2003). Physa acuta can tolerate polluted systems with high nutrient loads and low dissolved oxygen (Kalyoncu, Yıldırım et al., 2009) and thus, may survive in highly impacted systems. Therefore, we expect treatments with Physa sp. to reduce bio-film, stabilize ammonium, and reduce sediment by clearing settled algae.

Cladoceran filter feeders, such as D. magna, are important for several functions in aquatic systems. Daphnia spp. increase pH and available oxygen (Wojtal-Frankiewicz & Frankiewicz, 2011), alter disease risk (e.g., Kagami et al., 2004), and increase water clarity (Walsh, Carpenter & Van Der Zanden, 2016). They are important in nutrient cycling, specifically increasing nitrogen (N) or ammonium (NH4) in the water column and reducing P (Paterson et al., 2002; Mackay & Elser, 1998; Wojtal-Frankiewicz & Frankiewicz, 2011). These impacts may alter cyanobacterial competitive advantage (Mackay & Elser 1998, but see Paterson et al. 2002). However, changes in nutrient loads can in turn change the population dynamics of D. magna (Kleiven, Larsson & Hobæk, 1992; Sterner & Hessen, 1994). These feedback loops suggests that Daphnia spp. direction and size of response to a nutrient perturbation may not be obvious. Therefore, greater understanding of these feedback loops could lead to a better overall understanding of system level recovery. We particularly expect treatments with D. magna to reduce the population growth of algal communities, stabilize ammonium concentration, alter algal communities, and reduce sediment load as a result of grazing on algal cells before they settle.

We believe that small-scale experiments are a key piece of validating and understanding resilience theory and may even aid in discovering drivers applicable to particular systems. Our specific aim is to gain insights into whether having multiple types of feeding modes (i.e., more diverse traits in the system) increases a simple ecosystem’s ability to recover from perturbation. We approach this empirically by contrasting a simple experimental community of primary producers (algae) alone with communities that also include one or two consumers with distinct feeding modes. Since more functional traits allow for differential responses to perturbation we hypothesized that treatments with both functional feeding types will recover more completely (i.e., be more similar to an undisturbed system) than treatments with only one feeding mode type or with primary producers only. The focus of this experiment is thus to examine the differential effects of Physa sp. and D. magna on microcosm recovery that follows a eutrophication event. We expect more complete recovery if the species have differential responses to the perturbation.

Methods

Experimental set up

Experiments were run for 45 days in the greenhouse at McMaster University between January and March, 2019. Initiating all tanks in one day was impossible so we staggered starting tanks. We evenly distributed tank start dates across 10 days among treatments in order to avoid confounding start date and treatment. Tanks were organized randomly within the greenhouse on two tables, following a random number generator. We used Daphnia magna and Physa sp. as herbivores. Algae, predominantly Chlorella spp. (Fig. S1A), was the primary food source. Both algae and herbivores were from lab maintained populations.

Treatments:

Our experiment consisted of four herbivore treatments (Fig. 1): no herbivores, D. magna only, Physa sp. only, and both herbivore species. Tanks were either perturbed with a single eutrophication event or were not disturbed. Thus, we had a total of eight treatments. Each treatment had ten replicates, for a total of eighty tanks.

General tank set up

Tanks were filled with 1 L of water from the lab cultures of D. magna. Next, the algal mix was added until starting concentration of (mean ± sd) 14.5 ± 2.3 chlorophyll-a µg/L was reached. This concentration was chosen based on preliminary work where we found that smaller concentration sometimes led to complete green algal disappearance. Each tank with D. magna received twenty individuals larger than one mm. The number of D. magna used were based on preliminary work to determine how many D. magna were needed to start a colony that would be unlikely to be extirpated due to stochasticity or adjusting to new water. We searched for a balance between extirpation and not using so many individuals that we couldn’t seed all tanks. Each tank with Physa sp. received four individuals with an average size of (mean ± sd) 4.6 mm ± 1 mm. Following the introduction of the organisms, tanks were left undisturbed for twenty-one days (at a median temperature of 19 °C during the day, no shade) and monitored. Tanks were left open to allow for natural colonization of algae. Each tank had two Petri dishes placed at the bottom of the tank for capturing organic particle deposits. Tanks were stirred daily for aeration. After the twenty-one days, we added 100 mL of 1g/L concentration of a nutrient mixture (Miracle Gro, 15-30-15 parts of nitrogen, phosphorus, potassium, respectively) to tanks selected for the perturbation treatment. This concentration was used to ensure a detectable change in ammonium concentration of a great enough magnitude to be considered a different trophic state (sensu Kratzer & Brezonik, 1981).

Figure 1 Study design:Vector images courtesy of the Integration and Application Network (ian.umces.edu/media-library).

Data collection

As recovery from perturbation can only be inferred from measured state variables, we look at several different endpoints. First, we look at ecosystem level endpoints: seston settlement load, ammonium concentration and chlorophyll-a, as a proxy for algal concentration. We also examine population endpoints: filter feeder (D. magna) populations, grazer (Physa sp.) survival and reproduction, and algal community composition.

Ecosytem level effects

We measured seston sedimentation at the end of the experiment by removing Petri dishes and washing the particles into a centrifuge tube and then centrifuging for one minute to determine the volume of settled material.

We measured chlorophyll-a daily by taking a 3 ml homogenized sample of tank water and using a AquaFlor flourimeter to measure fluorescence of the sample. We measured temperature and pH daily with HACH Pocket Pro. Ammonium (NH4) was measured using a YSI Pro Plus twice a week.

Herbivore populations

Daphnia magna populations were estimated twice a week by counting four 10 mL samples of tank water from each tank. Daphnia magna were returned to the tank after counting. We recorded snail egg masses and juveniles as they appeared and at the end of the experiment. All deceased organisms remained in the tanks, allowing for natural decomposition.

Algal composition

We collected five mL water samples for algal identification at the start of the experiment, the day prior to perturbation (day 20) and at the end of the experiment (day 45). All algal samples were placed in a cooler for five to seven days before being taken to the lab to be analyzed. From each five mL sample we extracted a ten µl sub-sample, which was placed on a hematocytometer slide for counting. Using a Zeiss Primo Star compound microscope and the program Zen, we took pictures of each algal slide at 10x magnification. We took four pictures of each slide. Algae captured on each photo were manually counted and identified to the best of our ability using Manaaki Whenua Landcare Research algae guide (Manaaki Whenua-Landcare Research, 2014).

Analysis

All analyses were completed in the R statistical programming environment (R Core Team (2016) version 3.6.1). Data and code are publicly available at: https://github.com/jwerba14/Disturbance.

All of the following generalized linear mixed models were run using the lme4 package (Bates et al., 2015) and included a random effect of start date. A random effect of start date was included because even minor changes in conditions, for example, the light environment could have detectable effects on our endpoints.

Ecosystem level effects

For both chlorophyll-a and ammonium we ran linear mixed effect models with log-transformed data and an interaction term between herbivore treatment and perturbation. We used the emmeans or multcomp package to compare our a priori contrasts (Lenth, 2020; Hothorn, Bretz & Westfall, 2008). We aimed to assess the difference between disturbed and undisturbed treatments across herbivore treatments. For example, first, we evaluated differences between perturbation/no perturbation treatments for each D. magna and Physa sp. and then we asked whether the two species differ in their responses (differences in differences). We ran all six possible contrasts. Bonferonni adjustment for multiple contrasts was used to calculate p-values.

For final chlorophyll-a concentrations we used the last four days of the experiment. The maximum post-perturbation chlorophyll-a data used for analysis was a mean of values recorded in a three day window starting one day after the perturbation. For ammonium we ran the same model as for chlorophyll-a but took a single day maximum and the final day value because of the lower measurement frequency.

Herbivore populations

We examined the effect of perturbation on final populations of D. magna using a generalized linear model, with a negative binomial error distribution. We used disturbed (y/n) and herbivore treatment as predictors. Maximum populations were defined as the mean of all four sub-samples. We log transformed organisms’ max populations and used a weighting factor of 1variance of the sub-samples used to measure their abundance for the linear model. Population data may present an incomplete picture of D. magna response because of its ability to form dormant eggs. Therefore, we ran exploratory analyses to find out if the presence or absence of ephippia in D. magna at the end of the experiment were affected by perturbation and grazer presence (binomial generalized linear mixed model).

Physa sp. survival was modeled as the proportion surviving given day and treatment, using a generalized linear mixed model with a binomial error distribution and individual tank as a random effect. We used a semi-parametric bootstrap method to calculate confidence intervals within the lme4 package (Bates et al., 2015). The probability of snails laying eggs by the end of the experiment was also modeled with a binomial generalized linear model, with herbivore and perturbation treatment as fixed effects.

Algal composition

Algal community turnover between treatments and time were explored using PCoA on Bray-Curtis dissimilarity matrices. Variation explained by grazer treatment and time were analyzed using a permutational multivariate analysis of variance (PERMANOVA). These analyses were done in the Vegan 2.3.3 package (Oksanen et al., 2016).

Results

Chlorophyll-a concentrations

We directly test how herbivore treatments (none, Physa sp. alone, D. magna alone, or Physa sp. and D. magna in combination) affect the difference between disturbed systems and undisturbed systems. We test if the diverse herbivore treatment reduces the difference between the undisturbed and disturbed treatments more than the treatments with fewer types of herbivores. We find, for both final chlorophyll-a concentration (Fig. 2A) and maximum chlorophyll-a concentration (Fig. 2B) that in the presence of D. magna either alone or in combination with Physa sp., the difference between the undisturbed and disturbed treatments are smaller than when no herbivore is present (Table 1; for raw data, see Figs. 2C, 2D).

Ammonium concentrations

We are unable to detect any effect in the size of the difference between disturbed and undisturbed treatments for final (Fig. 3A) or maximum ammonium concentrations (Fig. 3B; Table 1; for raw data, see Figs. 3C, 3D).

Variable seston settlement

We find that D. magna alone increases the difference in seston settlement between disturbed and undisturbed treatments when compared to either treatment of no herbivore or only Physa present (Fig. 4A). But, D. magna appear to lower the raw variable seston settlement substantially in the disturbed treatments (Fig. 4B; Table 1 for model estimates).

Figure 2 Log10 fold change in final (A) and maximum (B) chlorophyll-a (µg/L) between undisturbed and disturbed treatments. Distance from 0 (the dashed vertical line) indicates the magnitude of the change. Raw data summary for final (C) and maximum (D) chlorophyll-a (µg/L).

Points are means and error bars are 95% CI. For all treatments N = 10.

Table 1 These are model estimates for each of our measures of resilience.

The estimate is difference between log10(disturbed)-log10(undisturbed) between herbivore treatments. The model column indicates the endpoint of interest. The contrast column indicates which treatments are being compared. For all treatments N = 10.

Model	Contrast	Estimate (log10)	SE	p	
Chlorophyll-a (µg/L) Final	None - Physa sp.	9.2	8.4	0.3	
	None-D. magna	33	12.3	0.02	
	None - both	33	12.3	0.02	
	Physa sp. - D. magna	24	9.19	0.02	
	Physa sp. - both	24	9.18	0.02	
	D. magna - both	−0.02	0.99	0.98	
Chlorophyll-a (µg/L)	None- Physa sp.	10.8	11.9	0.4	
Maximum	None-D. magna	60	12.1	0.0001	
	None - both	61	12.2	0.0001	
	Physa sp. - D. magna	49	10.6	0.0001	
	Physa sp. - both	50	10.7	0.0001	
	D. magna - both	0.97	1.7	0.58	
Ammonium (mg/L) Final	None - Physa sp.	1.05	1.5	0.5	
	None-D. magna	1.7	1.5	0.25	
	None - both	−0.31	1.5	0.8	
	Physa sp. - D. magna	0.64	1.4	0.65	
	Physa sp. - both	−1.36	1.4	0.35	
	D. magna - both	−2	1.3	0.14	
Ammonium (mg/L)	None-Physa sp.	−1.7	2.6	0.5	
Maximum	None- D. magna	−2.35	2.6	0.4	
	None - both	−4.6	2.85	0.12	
	Physa sp. - D. magna	−0.66	2.7	0.8	
	Physa sp. - both	−2.9	2.95	0.3	
	D. magna - both	−2.2	2.9	0.4	
seston settlement (mL)	None - Physa sp.	−0.24	0.3	0.5	
	None-D. magna	0.96	0.35	0.008	
	None - both	0.52	0.3	0.08	
	Physa sp. - D. magna	1.2	0.35	0.001	
	Physa sp. - both	0.765	0.3	0.01	
	D. magna - both	−0.44	0.3	0.56	

Figure 3 Log10 fold change in final (A) and maximum (B) ammonium (mg/L) between undisturbed and disturbed treatments. Distance from 0 (the dashed vertical line) indicates the magnitude of the change. Raw data summary for final (C) and maximum (D) ammonium (mg/L).

Points are means and error bars are 95% CI. For all treatments N = 10.

Figure 4 Log10 fold change in seston settlement between undisturbed and disturbed treatments (A) and treatment means (B) In (A) Distance from 0 (the dashed vertical line) indicates the magnitude to the change.

Points are means and error bars are 95% CI. For all treatments N = 10.

Daphnia magna reproduction and population

We see a clear change in reproductive strategy towards ephippia production in disturbed treatments (Fig. 5). And while we don’t see a downstream effect on ephippia production we do see a reduction in the final populations of D. magna populations when Physa are present (Fig. 6A). We cannot, however, detect a difference in the change between undisturbed and disturbed treatments in either final or maximum D. magna populations when Physa are or are not present (Table 2).

Figure 5 Ephippia presence at the end of the experiment (twenty-one days post perturbation).

Points are means and error bars are 95% CI. For all treatments N = 10.

Figure 6 Final (A) and maximum (B) D. magna populations per 10 mL water.

Points are means and error bars are 95% CI. For all treatments N = 10.

Physa sp.

Physa sp. in tanks without D. magna died more quickly than in tanks with D. magna (p = 0.004; Fig. 7). We were unable to detect differences in overall survival between treatments. Physa sp. egg production also increased when D. magna were present (Fig. 8). Additionally, when D. magna are present there is a larger difference in egg production between the disturbed and undisturbed treatments (Fig. 9; Table 3).

Algal communities

At the beginning of the experiment algal communities were indistinguishable between treatments (Fig. 10A). Directly before the perturbation there was some separation in algal community along the first PCoA axis between treatments with or without D. magna (Fig. 10B). Twenty-one days post-perturbation the treatments were clearly delineated into two groups, those with D. magna and those without. There is minor separation along the second PCoA axis between the disturbed and undisturbed treatments when D. magna is absent (Fig. 10C). PERMANOVA results suggest that herbivore treatment explained the most marginal variance in the algal community (R2 = 0.1, p =0 .0009), followed by time (which was modelled as a continuous variable) (R2 = 0.06, p =0 .0009). We are unable to distinguish between perturbation treatments, and the majority of variance in the community structure was not explained by any of our fixed effects (Residual Variance = 0.8).

Chlorella spp. made up the majority of the algal community for all treatments at all collection points (Fig. S1), followed by Scenedesmus spp. Very few other species ever represented more than 5% of the community. A full list of species found is provided in Supplemental Table 1.

Summary of findings

For the ecosystem level metrics (chlorophyll-a concentration, variable seston loads and ammonium concentration), neither functional group alone or their combination increased resilience for all three. In fact, while D. magna increased resilience in regards to chlorophyll-a concentrations (Fig. 2), their presence reduced resilience when seston sediment was factored in (Fig. 4). While the herbivore treatments, with or without perturbation, had clear and often strong effects on chlorophyll-a (Fig. 2), the effects on ammonium (Fig. 3) and seston settlement levels (Fig. 4) was less clear.

Discussion

We expect, by definition, that a more resilient system will return to the pre-disturbed state more fully than a less resilient system. Resilience is expected to be higher in more diverse systems (Schmitt et al., 2020; Bouska et al., 2019; Downing et al., 2012; Naeem & Li, 1997). However, we find, in contrast to our hypothesis, that having two different herbivore functional groups did not detectably reverse the effects of eutrophication.

Sediment loads are a problem following cultural eutrophication events (Kelly et al., 2018). Snails can reduce suspended particle concentrations (Mo et al., 2017). The fact that we could not detect the effect of Physa sp. on seston settlement loads may be due to the high snail mortality and low birth rates across treatments. Indeed, it was not until the end of our experiment that we began to see newborn snails and egg clutches. D. magna presence decreased the resilience of seston settlement as shown by the increased difference between perturbed and recovered system. At the same time, the disturbed systems with D. magna present showed reduced seston settlement compared to undisturbed systems suggesting that having a filter feeder, at least in low-diversity systems, may be critical to reducing the downstream effects of increased seston settlement (Elgin & Jackson, 2016). This reduction in seston settlement could be due to decreased carbonate precipitation through controlling algal biomass and decreasing both N and P sedimentation overall (Sarnelle, 1993). Why this occurs more in a disturbed system is unclear, though the reason may be tied to the increased population of D. magna in the experimentally disturbed systems.

Nitrogen loads are part of the cause of eutrophication. Both species of herbivore represented in our experiment are known to to be important in nitrogen cycling and likely both raise nitrogen concentrations (e.g., Paterson et al., 2002; Mulholland et al., 1991; Griffiths & Hill, 2014). Although the experimental snail populations may appear small, 1–3 snail per 1L of water represents a high density (Zimmermann, Luth & Esch, 2017). Thus, the effect of snails on nutrients may be consistent with expectations based on natural systems (Hall Jr, Tank & Dybdahl, 2003). Additionally, snails can alter nitrogen availability via selective grazing (Arango et al., 2009; Liess & Kahlert, 2009). It is possible that high mortality and decomposition of Physa sp. led to increased ammonium. This highlights a different pathway whose major impact is to change the timing of ammonium availability in water - another interesting complication likely to be associated with species diversity through the biodiversity sampling effect (e.g., Steiner et al., 2006). For this ecosystem-level response (ammonium concentration), presence of more than one herbivore functional group appears to deepen, at least initially, the effects of eutrophication on nitrogen load. The longer-term consequences of herbivores are unclear and are likely to have complex interactions with primary producers. Notably, no combination of herbivore species detectably altered the difference between disturbed and undisturbed treatments on ammonium concentration.

Ephippia production represents a different mode of reproduction for D. magna. Increased ephippia production may be due, at least partially, to the higher maximum and final populations of D. magna in disturbed treatments. D. magna produce ephippia when daylight is less than 12 hrs, at low food availability, when populations exceed 0.4 individuals per mL (Carvalho & Hughes, 1983), and when stressed by pollution (Ringot et al., 2018). We observe this shift in reproductive behavior far more frequently in disturbed treatments than in undisturbed treatments. We can rule out light as the cause because tanks shared a uniformly illuminated space. However, D. magna densities were much higher in perturbed treatments so this likely contributed to switching reproductive tactics, though Booksmythe et al. (2018) did not find increased density leading clearly to increased ephippia production. It is possible D. magna switch to ephippia production in new trophic states (i.e., change in available nutrients) because the ephippia are resilient to changes in trophic state (Isanta Navarro et al., 2019). If true, ephippia may signal a transition to a different state of the system. Alternative stable states often result from eutrophication (Carpenter, Ludwig & Brock, 1999).

Table 2 These are model estimates for D. magna populations.

The model column indicates the dependent variable of the model. The coefficient column indicates the fixed effect. The contrasts column indicates the specific contrasts we tested.

Model	Coefficient	Contrasts	Estimate	StDev	p	
Final populations		Disturbed-Undisturbed	1.04	0.2	<0.0001	
		D. magna-Both	−0.19	0.2	0.4	
	Interaction: Disturbed x Herbivore Treatment		−0.24	0.3	0.4	
Maximum populations		Disturbed-Undisturbed	0.6	0.2	0.005	
		D. magna-Both	−0.25	0.18	0.18	
	Interaction: Disturbed x Herbivore Treatment		−0.23	0.23	0.3	
Probability ephippia present		Disturbed-Undisturbed	−4.4	2.2	0.04	
		D. magna-Both	−0.48	1.4	0.7	
	Interaction: Disturbed x Herbivore Treatment		1.04	2.3	0.65	

Figure 7 Physa sp. survival over the course of the experiment.

Points are averaged mean daily survival across all replicates (N = 10 for all treatments); color (D. magna present: purple and blue lines; absent: green and red lines) and linetype (disturbed: solid; not disturbed: dashed) differentiate the four treatments. The envelopes around the fitted lines are 95% confidence intervals on Physa sp. survival over time.

Figure 8 Log-odds that Physa sp. laid eggs.

Points are means and error bars are 95% CI. For all treatments N = 10.

Figure 9 The change in likelihood of Physa sp. laying eggs between undisturbed and disturbed treatments.

Points are means and error bars are 95% CI. For all treatments N = 10.

Table 3 These are model estimates for Physa sp. populations.

The model column indicates the dependent variable of the model. The coefficient column indicates the fixed effect. The contrast column indicates the specific contrasts we tested.

Model	Coefficient	Contrasts	Estimate	StDev	p	
Survival		Physa sp.-Both	0.45	0.4	0.2	
		Disturbed-Undisturbed	0.19	0.4	0.6	
	Day		−0.09	0.006	<0.0001	
	Interaction: Day x Herbivore Treatment		−0.02	0.008	0.004	
	Interaction: Disturbed x Day		−0.003	0.007	0.7	
Probability eggmass present		Disturbed-Undisturbed	−2.05	0.55	0.0002	
		Physa sp.-Both	−4.9	0.7	<0.0001	
	Interaction: Disturbed x Herbivore Treatment		2.97	0.8	0.0002	

Figure 10 Algal community clusters.

Points represent the centroid of the algal communities. Error bars show standard deviation. Panels represent different time points: (A) starting structure, (B) mid-point, and (C) final day. Shapes indicate perturbation treatment: circles show no perturbation, triangles show disturbed treatments. Colors represent herbivore combination. Axis are PCoA 1 (x-axis) and PCoA 2 (y-axis). For all treatments N = 10.

Physa sp. were more likely to lay egg masses within the six weeks of our experiment if D. magna were present. As far as we are aware this interaction has not been observed elsewhere. Though a similar finding was found with Lymnea sp. which has been shown to increase fecundity in the presence of congeneric species, though other snails, though the mechanism remains unclear (Hershey, 1990). The combination of reduced D. magna population when Physa sp. is present but increased Physa sp. eggs (and therefore likely future increased population) when D. magna are present suggests a possible feedback loop. That is, increases in D. magna lead to increases in Physa sp. reproduction (eggs) which in turn eventually leads to increases in Physa sp. and, then likely, some impact of this larger Physa sp. population on D. magna. In nature, this may coincide with other factors contributing to D. magna reduction over the course of summer. This also highlights an advantage of our small-scale study in that we can see an interaction that may be hidden in more complex systems implying there are likely many under-detected interactions that nonetheless may have profound consequences on populations.

Our PERMANOVA results captured little of the variation found in algal compositions. This could be due to stochastic colonization (e.g., Kimbrel et al., 2019) as all our tanks were open to air in the greenhouse. Also, priority effects could account for much of the variability we observe. Small divergences starting populations can easily lead to large variation in community composition (Fukami, 2015).

Study Limitations

The goal of this study was to examine the effect of having two representative traits compared to one on resilience. In many ways this aim is a proof of concept that having a more diverse pool of a trait type (here two feeding modes instead of one) leads to increased resilience. This aim necessitated extreme simplification relative to a natural system. For example, we recognize that predation in natural systems plays a large role in structuring zooplankton communities (Ersoy et al., 2019) and that food webs have myriad sets of full ecosystem level effects (Knight et al., 2005); however, these are outside of the scope of this study. Additionally, in eutrophic systems cyanobacteria will often take over (ONeil et al., 2012), causing the system to move to a new cyanobacteria-dominated state instead of returning to a previous state (Viaroli et al., 2008).

We also encountered difficulties with sampling and identifiying the algal community. First, we only sampled a small percentage of the tanks for algae due to time constraints; and second, the algal community was challenging to identify for many of the reasons presented in Manoylov (2014). As such, many species as used in our PCOA were characterized simply as unknown species X (based on morphological differences). If these were in fact different species, this would produce no statistical issue (just an unsatisfying biological issue of having unidentified species). However, some species come in different morphs, which could cause us to inflate diversity and misunderstand the community dynamics. We also cannot comment on specific algal species contributions to nutrient dynamics. We therefore suggest caution when interpreting our results about the algal community; however, our results do concur with what is expected from the literature. The differences in the algal community were mostly driven by herbivore treatments. This effect is not surprising as herbivores influence algal communities through direct and indirect routes (Ger et al., 2019; Abrantes et al., 2006; Sterner, 1989).

Finally, our experiment was only six weeks long. We mention this to highlight that, although some experimental results are informative, long-term consequences of interactions among the components may be confounded by processes unfolding at different time scales. This applies to dynamics observed under lab conditions, and likely even more so in natural situations.

Future directions

More direct experiments that decouple functional diversity and biodiversity are critical for moving forward with our understanding of ecosystem resilience. For example, studies that can increase richness without increasing functional types would help separate functional trait effect from richness effect (e.g., two cladocerans compared to a cladoceran and a snail species). These types of experiments also have the advantage of potentially highlighting combined or opposing effects of species (e.g., Mo et al., 2017) that may be difficult to disentangle in a natural system.

Conclusions

Overall, we have tentative evidence that functional types have combined effects on the system but we do not see that translate to resilience across measures. Measurements of resilience can only be done via signature variables thus the choice of end points may greatly influence the outcome. We are unable to comment on long-term complex interactions such as continued ammonium availability and shifts in algal communities and edibility. Our study suggests that resilience may not be predictable by a simplified approach that uses functional traits. Instead, it involves complex interactions that require knowledge of adaptive species responses and indirect effects. Furthermore, these complex interactions appear strongly context-dependent.

Supplemental Information

Supplemental Information 1 Algal composition across treatment types and time

Click here for additional data file.

We would like to thank Josephine Huynh and Adit Chokshi for help with data collection, Dr. Susan Dudley for access to McMaster Greenhouse and Dr. Ben Bolker for help with data analysis.

Additional Information and Declarations

Competing Interests

Author Contributions

Data Availability

The authors declare there are no competing interests

Jo A. Werba conceived and designed the experiments, performed the experiments, analyzed the data, prepared figures and/or tables, authored or reviewed drafts of the article, and approved the final draft.

Alexander C. Phong conceived and designed the experiments, performed the experiments, authored or reviewed drafts of the article, and approved the final draft.

Lakhdeep Brar performed the experiments, authored or reviewed drafts of the article, and approved the final draft.

Acacia Frempong-Manso conceived and designed the experiments, performed the experiments, authored or reviewed drafts of the article, and approved the final draft.

Ofure Vanessa Oware performed the experiments, authored or reviewed drafts of the article, and approved the final draft.

Jurek Kolasa conceived and designed the experiments, prepared figures and/or tables, authored or reviewed drafts of the article, and approved the final draft.

The following information was supplied regarding data availability:

The data and code are available at GitHub: https://github.com/jwerba14/Disturbance.

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
