# Peer review of "Interactions between two functionally distinct aquatic invertebrate herbivores complicate ecosystem- and population-level resilience"

_PeerJ, doi:10.7717/peerj.14103_

## Round 0.1 · original submission · Major Revisions

· Academic Editor

Major Revisions

Dear Dr. Werba,

After the first review round, your manuscript reached all kinds of statuses (Rejection, Major, and Minor Revisions). Still, I think that you may be able to improve your manuscript in a manner that pleases all the reviewers. Therefore, I am granting you a major review in order for you to take care of all the issues raised by the reviewers.

Sincerely,
Daniel Silva

·

Basic reporting

Thank you for giving me the opportunity to read the paper “Interactions between two functionally distinct aquatic invertebrate herbivores complicate ecosystem- and population-level resilience” by Werba et al. In this paper, the authors report on the results of an experimental study to examine the differential effects of Physa sp. and Daphnia magna on microcosm recovery that follows a eutrophication event. I appreciate the time and effort that went into this study. Even so, it is my professional opinion that the paper is not publishable in its current form or context. The main reason I feel the paper is not publishable is that it needs clarification in a number of places, and I do not completely agree with all aspects of the conclusions of the paper, which in my opinion are not supported by the data and analyses. Moreover, the manuscript is not well organized and clear, and it is quite long for the amount of data presented. Finally, some critical data and context is missing, including the phytoplankton composition in the study, and many classic references are missing. Overall, these limitations reduce the logical flow, obscure the essence and relevance of the conclusions, and diminish the potential impact for a wider range of readers.

Experimental design

Comment 1: The Methods section of a manuscript should be sufficiently detailed so that the reviewer and the readership have a clear understanding of: (1) how the data were collected or obtained, (2) how the key variables were defined and measured, (3) the study design, and (4) the statistical methods used for data analysis. In my professional opinion, the authors failed in points 1, 2, and 3. To be more clear, the experimental set-up subsection (2.1) should be divided into more subsections (2.1.1, 2.1.2, 2.1.3, etc.) and each step should be better described. The authors did in the "Analysis" section. I had a hard time trying to understand the experiment, and even after a while, I do not know if a fully understood. I strongly suggest a figure showing the experimental design.

Validity of the findings

Comment 2: The results should be presented with a neutral description and without questions, explanation, or interpretation, as the authors did in lines 164-168 and in line 183. Research questions and explanations about the analyses carried out are to be given in the Introduction and in the Methods section, respectively, whereas interpretations as to the importance and meaning of the results are to be given in the Discussion section.

Comment 3: In my opinion, one weakness of the results is that the authors did not provide the composition of the phytoplankton community. Cyanobacteria is commonly found in phytoplankton communities during a eutrophication event, and its presence can shift the composition of the herbivore community, and even the importance of functional traits (https://doi.org/10.1007/s10750-018-3710-0). Could the authors provide a list of phytoplankton species found in the experiment?

Comment 4: Furthermore, cyanobacterial dominance may also regulate the herbivores via bottom-up effects. The authors only consider the top-down effect of herbivores on phytoplankton, but not the other way around. What do the authors think about this aspect and whether it should be incorporated as a possible explanation of the results?

Comment 5: According to Docherty and Smith (1999; DOI: 10.1136/bmj.318.7193.1224), a discussion should be prepared by organizing information in the following order: (a) statement of principal findings; (b) strengths and weaknesses of the study; (c) strengths and weaknesses in relation to other studies, discussing particularly any differences in results; (d) meaning of the study: possible mechanisms and implications; (e) unanswered questions and future research. In my professional opinion, the authors failed to provide all the information mentioned above, especially the strengths and weaknesses of their study and in relation to other studies, as well as possible mechanisms and implications of their findings.

Additional comments

Comment 6: There are significant errors in some of the Figures and also a lot of missing information including axis labels, units, etc.

Comment 7: The authors choose a new style of figure caption, which include the following elements: a declarative title that summarizes the result or major finding of the data that are being presented in the figure, a brief description of the methods necessary to understand the figure without having to refer to the main text, and statistical information. However, not all figures contain all these elements, which makes it difficult for readers to understand the style. As it is an unusual style, I recommend presenting all the elements in all captions in the right order or choosing the "traditional" way, in which the captions did not overtly state the main finding of the data being presented.

Comment 8: The captions should be presented in the same way as described above.

Comment 9: The authors referenced figures in the entire Discussion section. The most common convention is that tables and figures are not referenced from the Discussion section of a paper. I strongly recommend the authors do not repeat detailed results that can be found in the Results section. In general, specific figure numbers do not need to be re-stated in the Discussion unless you feel that doing so would substantially enhance your argument or discussion point. In my opinion, this is not the case.

Reviewer 2 ·

Basic reporting

Particularly I liked the beginning of introduction, giving general concepts and refining into next topic. However, I would like to suggest to add some reference about functional groups or functional ecology, few phrases are welcome.

On English language - there are some parts that can be reduced or simplified.

Title suggestion: ''Increasing ecosystem complexity and population-level resilience by interactions between two functionally different aquatic invertebrates''
-If both are herbivorous, they are not functionally similar?

Please see more suggestions in the attached file.

Experimental design

Please see the attached file.

Validity of the findings

Please see the attached file.

Additional comments

Please see the attached file.

Annotated reviews are not available for download in order to protect the identity of reviewers who chose to remain anonymous.

Reviewer 3 ·

Basic reporting

The manuscript has clear and objective writing. The introduction brought by the authors shows well the content studied and the questions that seek answers, in addition the literature is mostly current and relevant. The structure of the article complies with the required standards as well as the data presented. The figures are clear and have high resolution, but I suggest some changes in the comments below.

Experimental design

The research is within the scope of the journal. The methodology used is well described and the experimental design presented must have demanded a lot of work from the authors.

Validity of the findings

The data obtained by the authors are interesting and valid, however, they can be better explored. The novelty of the work is guided by the difficulty in measuring the resilience of environments in addition to the contradiction of some definitions. What makes this question important to be investigated by researchers. The conclusion summarizes well what the authors found, however, it can also be improved.

Additional comments

Abstract: I suggest highlighting in the first lines why it is important to study about resilience, what is the relevance of the work (similar to the beginning of the introduction), to better contextualize.

Introduction
Lines 56-58: put the informatoin in one phrase and avoid repeat words.
Line 66. Put in () which means the abbreviations since they were not mentioned before. The ammonium was mentioned before, but they didn't put the formula between ().

Metodology
Lines 101 and 103: Standardize the presentation of results with standard deviation, I suggest omitting the sd, like: 14.5 ± 2.3 μg/L of chlorophyll-a.
Line 103: Why were the tanks undisturbed for 21 days, for the organisms to adapt to the medium?
Line 109: add the comma after the parentheses (AquaFlor fluorometer)

Results
Lines 167-168: I suggest removing this question from the results session.
Line 171: I suggest citing the figure after the result to which it belongs, not putting them all at the end of the sentence.
Line 172 – 173: Leave this assumption for the discussion session, as in the presentation of all other results the authors do not suggest what the results might indicate.
Line 197: the authors identified the algae species only in the supplementary material, I suggest highlighting the species found.

Discussion
Discussion section – the first three paragraphs of this section are descriptive and without references to the literature. It looks like a quote from the results. Add these paragraphs to the results, or bring data from the literature and turn it into a discussion, or simply remove it from the manuscript.
Lines 213 e 215: When the authors cite 'algal concentration' what do they mean? The amount of algal cells found, that is, all the cells that were in the community? Or was the concentration of algae defined by the concentration of chlorophyll a?
Line 215: Is the Figure cited 1 AND 2? It is missing an “and” between the numbers and also closing the parenthesis.
Line 226: Can the authors attribute any factor to the death of the snails? Were the animals sensitive to laboratory cultivation? Even if the animals have died, does the validity of the experiment remain?
Line 289: Is there a punctuation ')' in the middle of the sentence, is it a typo, or is the sentence missing a parenthesis?
Line 291. I suggest citing all species found.

Conclusion
Line 304: The term synergistic effects appears only in the conclusion and not in the rest of the entire text – I suggest further explaining the term at some point in the manuscript.
Line 306: Repetition of “long-term”. Replace this words.
I suggest that the authors improve the conclusion, making evident the difficulties encountered in measuring resilience and the conflict between the parameters that are measured. Evidencing many points that the authors are unable to infer any interpretation reduces the importance of the manuscript.

Figuras
Figures: All figures contain explanations that are not required in the legend. The caption needs to be clear and objective so that the reader understands the figure. It is not necessary to interpret results in the legend.
Example: in figure 1 “Both treatments containing D. magna reduced the proportional change between disturbed and undisturbed treatments when compared to Physa alone or to no herbivore.” – This is not necessary on legend.
In addition, I suggest that the authors assemble the figures, grouping them into one, two or more, and thus adding some figures from the supplementary material to further explore and discuss the results.

Supplementary material
How do the authors explain the size of error bars? In some figures, some measurements have very large error bars.
Whats means ‘y’ and n on figure S1?

---

## Round 0.2 · Minor Revisions

· Academic Editor

Minor Revisions

Dear Dr. Werba,

After a new review round. The reviewers believe your manuscript has improved a lot. Therefore, your manuscript received a minor review. Congratulations.

Daniel Silva

·

Basic reporting

Thank you for the new version of the manuscript #70031. The authors did a good job incorporating the suggestions made in the previous revision and the manuscript reads much better as a result. I saw that they also incorporated the other reviewers’ suggestions. As a result, I believe the manuscript has greatly improved in its quality and potential impact and is now ready for publication after a final editorial check. I only have very minor edits below. In general, the writing style should be clearer and more concise, particularly the Introduction section which in some parts is cumbersome and repeatable. Some examples are given in the specific comments. Here, I would emphasize that all suggestions are easily doable.

Experimental design

Everything is OK here.

Validity of the findings

Everything is OK here.

Additional comments

Specific comments
1. Line 14: What do the authors mean by "prolific work", creative works or many works? Also, Isn't "prolific works", in its plural form? The authors should consider another word to use in the place of "prolific", such as "many works" or "plenty of works" whether the meaning is related to quantity.

2. Line 16: Please reformulate the sentence “tests of these ideas have yielded mixed results”. Suggestions: "the validation of these ideas have yielded mixed results" or "works that tested these ideas have yielded mixed results".

3. Lines 15-18: What is mediated by "more complex pathways"? Resilience? If so, maybe a connector such as "and" between the sentences is missing here, which obscures the text flow. The logic is: "There is some evidence that resilience increases with biodiversity" and there is also evidence that "it (resilience) is mediated through more complex pathways. Please, reformulate this sentence.

4. Line 18: Please, use "dominant" in the place of "critical".

5. Lines 18-20: It is hard to understand the authors' point here. Moreover, I observed that the authors have been repeating the same word many times throughout the text. Here, the authors repeated the word "example" three times. I suggest reformulating this sentence. Suggestion: "However, counter-examples are also common for each work that supports a particular mechanism that increases resilience."

6. Lines 37-38: Is there a reference for this sentence?

7. Line 52: Please, remove this repeated word.

8. Lines 96-107: The authors should make clear in the text that nutrient amounts and the number of Daphnia to start the experiment were based on preliminarily work (the same explanation given in the rebuttal letter). This is an important step in laboratory experiments (preliminarily work) and a detailed explanation helps reproducibility of the experimental set-up.

9. Supplementary material: The authors have added a table of the phytoplankton list to supplement table 1. In my opinion, the table is not very informative given the large number of unknown species on the list. I suggest grouping each unknown species in one category (I named “unknown species”) and presenting the phytoplankton composition in a figure format instead of a table. My suggestion is based on the premise that what matters in this result is the contribution (%) of unknown species to the total identified species, as well as the identified species that most contributed to the composition. I made some figures based on the data provided in the supplementary file, and the authors can see them attached to the review. As I elaborated these figures in R, I am also providing the script file together with the dataset that I elaborated with the data provided by the authors. Feel free to use the figures or at least the idea. If the authors wish to keep the table format (which I strongly do not recommend) I suggest reviewing the name “unknown” in the table that sometimes appeared as “known”.

Reviewer 2 ·

Basic reporting

The authors improved the manuscript according to our suggestions.
I recommend the acceptance.

Experimental design

It was improved according our suggestions.

Validity of the findings

This is an important manuscript, clear, direct and well written.

Reviewer 3 ·

Basic reporting

No comment.

Experimental design

No comment.

Validity of the findings

No comment.

Additional comments

I would like to thank you for the opportunity to read the paper ”Interactions between two functionally distinct aquatic invertebrate herbivores complicate ecosystem- and population-level resilience.”.

The authors made all the corrections proposed by the reviewers and, in general, the work improved a lot. There are new references in the introduction and a good explanation of the work; the methodology section with the new division of steps was well organized; the results were well described and the discussion was interesting with a good basis in the literature. Therefore, I congratulate the authors and collaborators for their work.

The only suggestion I would still make, just for visual aesthetics, is the abbreviation of treatments on the “y” axis of the graphs instead of writing what each thing means in full (this could be explained in the legend, so that the graph would be without so many writings).

---

## Round 0.3 · accepted · Accept

· Academic Editor

Accept

Dear Dr. Werba,

Congratulation! I am pleased to inform you that your manuscript has been formally accepted for publication in PeerJ.

Sincerely,
Daniel Silva, PhD

·

Basic reporting

Thank you for the new version of the manuscript #70031. The authors did a good job incorporating minor suggestions, especially the supplementary figures. As a result, I believe the manuscript is now ready for publication.

Experimental design

No comment.

Validity of the findings

No comment.

Additional comments

No comment.